# Charge density wave surface reconstruction in a van der Waals layered material

Sung-Hoon Lee [1] ✉ & Doohee Cho [2] ✉

Surface reconstruction plays a vital role in determining the surface electronic structure and chemistry of semiconductors and metal oxides. However, it has been commonly believed that surface reconstruction does not occur in van der Waals layered materials, as they do not undergo significant bond breaking during surface formation. In this study, we present evidence that charge density wave (CDW) order in these materials can, in fact, cause CDW surface reconstruction through interlayer coupling. Using density functional theory calculations on the 1T-TaS$_2$ surface, we reveal that CDW reconstruction, involving concerted small atomic displacements in the subsurface layer, results in a significant modification of the surface electronic structure, transforming it from a Mott insulator to a band insulator. This new form of surface reconstruction explains several previously unexplained observations on the 1T-TaS$_2$ surface and has important implications for interpreting surface phenomena in CDW-ordered layered materials.

The transition metal dichalcogenide 1T-TaS$_2$ has been an exclusive material platform for investigating the charge density wave (CDW) order and strong electron correlation in two dimensions (2D)[1–11]. Recently, it has become the focus of attention due to the competition between in-plane electron correlation and interlayer coupling of CDWs[12–24]. The triangular lattice of Ta ions in the $d^1$ electronic configuration that makes up a 1T-TaS$_2$ layer triggers CDW instabilities. When cooled to below 180 K, the Star of David (SoD) CDW distortion with a $(\sqrt{13} \times \sqrt{13})$ periodicity is observed. While the 12 Ta ions in the enlarged unit cell form a CDW gap, the remaining Ta ion at the center of the SoD creates a half-filled state. These half-filled localized states in a single layer result in a 2D Mott insulator[2,25,26] with a peak-to-peak gap of 0.5 eV between the Hubbard bands[27,28]. Bulk 1T-TaS$_2$ is also an insulator with a bandgap of 0.1 eV[29] and was previously thought to be a Mott insulator with negligible interlayer coupling[2–11]. However, recent density functional theory (DFT) calculations[12–15] have shown that interlayer coupling is substantial, and CDW stacking along the out-of-plane direction has a significant impact on the electronic structure. One particular bilayer CDW stacking configuration, called *AL* stacking, has been identified as the most stable[15] and has been confirmed by various experimental studies[16–18,30–33]. This *AL* stacking causes interlayer dimerization between the localized states, making the system a band insulator[14,15].

Studies of the 1T-TaS$_2$ surface have uncovered a more intricate interaction between electron correlation and CDW stacking. Scanning tunneling microscopy (STM) studies[19–21] have identified three different surfaces: two insulating surfaces with peak-to-peak gaps of 0.4 and 0.24 eV and metallic surfaces. The difference was attributed to variations in CDW stacking on the surface[19]. Taking into account that bilayer CDW stacking permits two potential cleavage planes, the large-gap insulating surface was associated with the bilayer-terminated surface, while the small-gap insulating surface was linked to the single-layer-terminated surface. Although a many-body theoretical study[22] provided support for these assignments, a subsequent STM study[23] raised an issue with this understanding. Both the upper and lower terraces of a single-layer surface step exhibited the same large gap, despite one of the two terraces having to be single-layer-terminated in the case of bilayer CDW stacking. To reconcile this inconsistency, it was suggested that the large gap is also a single-layer feature and a correlation-driven Mott gap[23]. On the other hand, angle-resolved photoemission spectroscopy (ARPES) data[8,34] showed two bands near the Fermi level with distinct $k_z$ dispersions: a dispersive band due to interlayer coupling[15,35] and a flat band at around −0.2 eV. Given that the peak at −0.2 eV originates from a large-gap surface with peaks at ±0.2 eV[19–21], the flatness of this peak along the $k_z$ direction remains unexplained.

[1]Department of Applied Physics, Kyung Hee University, Yongin, Republic of Korea. [2]Department of Physics, Yonsei University, Seoul, Republic of Korea.
✉e-mail: lsh@khu.ac.kr; dooheecho@yonsei.ac.kr

Here, using density functional calculations, we investigate the interplay between electron correlation and interlayer coupling on the surface of CDW-bilayer-stacked 1T-TaS$_2$. Our investigation reveals a previously unknown surface reconstruction, which explains the puzzling observations on the surface. Specifically, we determine the CDW structures of the large-gap, small-gap, and metallic surfaces, including their respective energy stabilities and spin structures. The large-gap surface is characterized by a band-insulating bilayer-terminated surface, while the small-gap surface is a Mott-insulating single-layer-terminated surface, which interestingly is less stable than the former. Consequently, when the area of the unstable small-gap surface is large, a spontaneous CDW reconstruction takes place, with the Mott-insulating single layer shifting into the third layer and leaving a CDW bilayer on the surface. The resulting two types of bilayer-terminated surfaces—one with a subsurface 2D Mott insulator and the other without—account for both the STM observation at surface steps and the flat ARPES band. In addition, we identify metallic surfaces as Mott insulators that can be described by trilayer Hubbard models. Overall, our results provide a firm basis for understanding the complex interaction between electron correlation and interlayer coupling in 1T-TaS$_2$. The CDW surface reconstruction we propose involves modest atomic displacements, yet it causes significant changes in the surface electronic structure and is highly likely to occur in other CDW-ordered layered materials as well.

## Results

In this study, we adopt the CDW stacking notation from ref. 15. Due to the three-fold rotational symmetry of the SoD cluster (Fig. 1a), it has five distinct stacking interfaces denoted by stacking vectors $\mathbf{T}_s = \mathbf{c}$, $\mathbf{a} + \mathbf{c}$, $2\mathbf{a} + \mathbf{c}$, $-2\mathbf{a} + \mathbf{c}$, and $-\mathbf{a} + \mathbf{c}$. These interfaces are labeled as $A$, $B$, $C$, $L$, and $M$ and are written in italics (refer to Fig. 1c, d). It is worth noting that the $C$ and $L$ interfaces are different when considering the S sublattice, which is not displayed in Fig. 1c.

### Determination of $U$ for surface calculations

Prior to examining surface properties, we assess the stability of bulk 1T-TaS$_2$ with respect to CDW stacking and the Hubbard $U$ potential to determine the appropriate range of $U$ values in our DFT + $U$ calculations (Fig. 2). For nonmagnetic cases, $L$ stacking is found to be the most stable among the five single-layer stacking configurations across all considered $U$ values. Moreover, the bilayer $AL$ stacking exhibits even greater stability than $L$ stacking for all $U$ values. This result corroborates and expands upon a previous finding[15], with $U = 0$ eV, that the ground-state CDW configuration for 1T-TaS$_2$ is $AL$ stacking when nonmagnetic. The high stability of the $L$ interface originates from two factors: enhanced van der Waals energy gain and favorable interlayer S–S bonding, with $AL$ stacking further increasing stability through a bandgap opening and accompanying electronic energy gain[15]. (The screw variant of $AL$ stacking[33], with a unit stacking sequence of $ALAIAH$, is nearly degenerate with the sliding stacking, showing an energy difference of less than 0.5 meV/SoD. Given this negligible difference, the present study employs $AL$ stacking as the representative case.) When magnetic order is considered, $L$ stacking begins to favor layer-by-layer antiferromagnetic (AFM) order at $U \approx 1.4$ eV and becomes the most stable stacking configuration above $U_c = 1.92$ eV. The onset of layer-by-layer AFM order for $A$ and $AL$ stacking is delayed to $U \approx 2.5$ and 2.1 eV, respectively, due to stronger interlayer coupling at the $A$ interface. Consequently, the ground-state CDW stacking phase is nonmagnetic $AL$ stacking for $U < U_c$ and layer-by-layer AFM-ordered $L$ stacking for $U > U_c$. The experimental evidence for bilayer $AL$ stacking[16–18,30–33] sets a limit of $U < U_c$ in our DFT + $U$ calculations. Our calculations suggest that the $U$ values of 2.27–2.94 eV calculated in previous DFT + $U$ studies[12,36,37], obtained using Dudarev et al.'s scheme[38] for Ta 5$d$ orbitals in 1T-TaS$_2$, are too high and may overestimate electron correlation. We employ a moderate $U$ value of 1.25 eV in our subsequent surface calculations.

### Large-gap and small-gap insulating surfaces

We begin by investigating two basic surfaces of 1T-TaS$_2$: one cleaved between bilayers and the other within a bilayer. We model these surfaces using eight- and seven-layer slabs, respectively, both with a ground-state $AL$ stacking. The eight-layer slab, with bilayer termination on both surfaces, is a band insulator that lacks any magnetic order; it does not exhibit magnetic order, even with $U = 1.75$ eV. The bandgap of

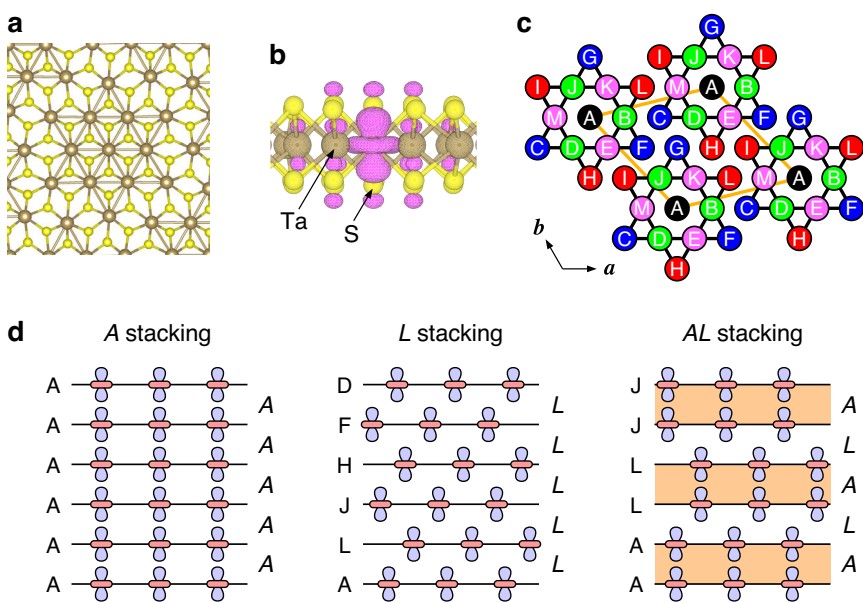

**Fig. 1 | Star of David (SoD) distortion and CDW stacking order in 1T-TaS$_2$.**
**a** Atomic structure of 1T-TaS$_2$ under the SoD distortion. **b** Isodensity surface of the half-filled state that is localized at the center of the SoD cluster. **c** Labeling convention for Ta atoms in an SoD cluster (from ref. 15). **d** Arrangement of half-filled states for $A$, $L$, and $AL$ stacking, with markings indicating the location of the SoD's center for each layer on the left and the CDW stacking interface between the layers on the right (in italics).

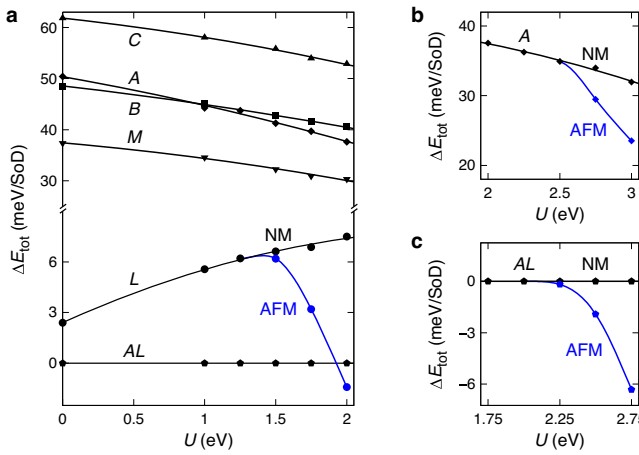

**Fig. 2 | Stability of bulk 1T-TaS₂ as a function of CDW stacking and the Hubbard U potential. a** Energy stability of five single-layer stackings and one bilayer stacking. For $L$ stacking, the stability of the layer-by-layer antiferromagnetic (AFM) order (blue) is compared to that of the nonmagnetic (NM) case (black). **b, c** Onsets of layer-by-layer AFM order in $A$ stacking and $AL$ stacking, respectively.

the slab is determined by the layers inside, and it is calculated to be 0.06 eV, which is nearly equivalent to the bulk bandgap of 0.05 eV. The spectral weight of the Ta $5d_{z^2}$ orbital at the SoD's center in the top layer, as shown by the shaded area in cyan in Fig. 3a, indicates that the surface bilayer has a slightly increased interlayer-hybridization gap due to minor surface relaxation. The corresponding projected density of states (PDOS) manifests two peaks at −0.18 and 0.08 eV, exhibiting a peak-to-peak gap of 0.26 eV. Considering that DFT calculations typically underestimate the bandgap substantially, the peak positions and intensities of the calculated PDOS align well with the STM $dI/dV$ spectrum of a large-gap (0.4 eV) insulating surface (Fig. 3c, blue lines). This finding implies that the peaks at ±0.2 eV identified by STM and ARPES correspond to the bonding and antibonding hybridization bands of the surface bilayer.

The seven-layer slab with an undimerized top layer exhibits a narrow half-filled band in the nonmagnetic calculation (Fig. 3b). Upon inclusion of spin polarization, the half-filled band becomes fully gapped, causing the split Hubbard bands to merge with the bulk bands (Fig. 3b). The calculated PDOS of the Ta $5d_{z^2}$ orbital in the top layer displays a peak-to-peak gap of 0.16 eV between the upper and lower Hubbard bands, as well as a double-peak structure in the occupied states. These PDOS features are well represented in the STM spectrum of a small-gap (0.24 eV) insulating surface (Fig. 3c, red lines). The spatial distribution of spin indicates that 88% of the spin is confined to the top layer (Fig. 3b). These results indicate that the single CDW layer on the surface behaves as a 2D Mott insulator that is decoupled from the bulk.

### Correlated surface trilayers and "metallic" surfaces
The extent of decoupling between the surface single layer and subsurface layers depends on the interlayer hopping constant ($t_\perp$), which is determined by the stacking interface between them. The value of $t_\perp$ is the largest at the $A$ interface ($\mathbf{T}_s = \mathbf{c}$), while it is the smallest at the $L$ or $C$ interface ($\mathbf{T}_s = \mp 2\mathbf{a} + \mathbf{c}$). It is instructive to consider the strongly coupled case of $A$-interface termination, wherein a surface trilayer is formed (Fig. 4a). Even in this case, the band structure suggests that the surface acts as a Mott insulator. Notably, the spin of the occupied Hubbard band, in this case, is divided equally between the first and third layers, bypassing the second layer. This behavior can be explained by regarding the surface trilayer as a 2D lattice of vertical trimers with small lateral hopping constants. A single trimer with equal nearest-neighbor hopping integrals of $t$ has three eigenstates with

eigenvalues of $\pm\sqrt{2}t$ and 0, and the zero-energy eigenstate is (1, 0, −1) (see Supplementary Note 1). The half-filled zero-energy eigenstates of the vertical trimers form the half-filled band of the surface trilayer, as evidenced by the nonmagnetic band structure and PDOS in Fig. 4c. Furthermore, the nonmagnetic band structure illustrates that the energy level corresponding to $-\sqrt{2}t$ is −0.22 eV, indicating that the interlayer hopping constant at the $A$ interface, $t_\perp^A$, is 0.16 eV. It also shows that the bandwidth of the half-filled band is only 0.07 eV, implying that the in-plane hopping constant ($t_\parallel$) is much smaller than $t_\perp^A$. When spin polarization is switched on, the half-filled band splits into upper and lower Hubbard bands with a peak-to-peak gap of 0.07 eV (Fig. 4a). The large decrease in the gap compared to that of the $L$-interface single-layer surface (0.16 eV) is caused by the doubled spatial extent of the half-filled states.

The experimental studies using STM have suggested that the single-layer surfaces with the $M$ or $B$ interface ($\mathbf{T}_s = \mp\mathbf{a} + \mathbf{c}$) exhibit "metallic" spectra[19,23]. The surface electronic structure of the $M$-interface case (Fig. 4b) lies between those of the $L$- and $A$-interface cases: the three surface layers form a 2D correlated system, with asymmetrically split spin in the first (71%) and third (26%) layers of the surface trilayer. The trimer model with asymmetric nearest-neighbor hopping integrals $t_\perp^M$ and $t_\perp^A$ yields an estimate of 0.61 for the ratio $t_\perp^M/t_\perp^A$ (Supplementary Note 1), indicating that $t_\perp^M$ is around 0.10 eV. The calculated top-layer PDOS shows good agreement with the STM $dI/dV$ spectrum of a "metallic" surface, as shown in Supplementary Fig. 1. It is important to note that the split Hubbard bands of the surface trilayer, with a peak-to-peak gap of 0.09 eV, and the absence of a metallic peak at the Fermi level in the STM spectrum indicate that the "metallic" surface is not a metal but instead a Mott insulator. The single-layer surfaces with the $B$ and $C$ interfaces exhibit similar characteristics to those with the $M$ and $L$ interfaces, respectively (Supplementary Fig. 2).

### Surface energetics
In Fig. 5a, we compare the surface energies ($E_{surf}$) for the various surfaces we have studied. The surfaces can be categorized into three groups based on their stability: the bilayer-terminated surface is the most stable with $E_{surf} = 2.31$ eV/($\sqrt{13} \times \sqrt{13}$); the single-layer surface of the $L$ interface is less stable by ∼ 30 meV/($\sqrt{13} \times \sqrt{13}$); and all the other single-layer surfaces, including the trilayer ($A$-interface) case, are even less stable by over 40 meV/($\sqrt{13} \times \sqrt{13}$). These results are consistent with the energetics for CDW stacking in the bulk, as shown in Fig. 2, and provide a rationale for observations from STM studies. Previous research[7-10,19-21,23] has shown that most surfaces, whether cooled to or cleaved at cryogenic temperatures, exhibit a large gap. This observation aligns with the notion that the bilayer-terminated surface is the most stable. The small-gap insulating surface, identified as the $L$-interface single-layer surface, has predominantly been observed in small domains near step edges or domain boundaries[19,23], unless stabilized by alkali metal adatoms[21], indicating its inherent instability. The "metallic" surfaces appear less frequently than the small-gap surfaces[23], underscoring their higher instability. The presence of these unstable single-layer surfaces may stem from the energy released during the cleaving process, estimated to be on the order of 1 eV/($\sqrt{13} \times \sqrt{13}$). On the whole, the calculated surface energies provide valuable insights into the stability and features of the different surfaces.

### CDW surface reconstruction
The STM observations of a large gap in both the upper and lower terraces of single-layer steps[23] can be explained by the presence of a buried single layer. If a single CDW layer on the surface is covered by a CDW bilayer with the $L$ stacking interface (Fig. 5b), the surface energy is lowered by 21 meV/($\sqrt{13} \times \sqrt{13}$), which brings it considerably close to the energy of a plain bilayer-terminated surface without a subsurface single layer (Fig. 5a). Interestingly, despite the single layer buried

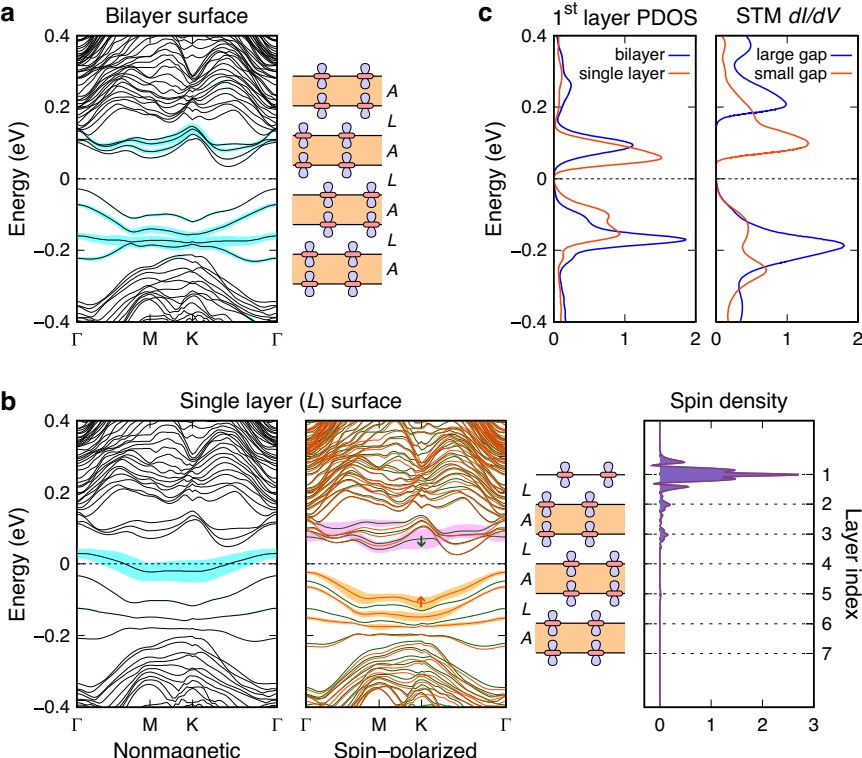

**Fig. 3 | Electronic structures of the large-gap and small-gap insulating surfaces.** **a** Nonmagnetic band structure of an eight-layer slab with *AL* stacking, where both surfaces are terminated with CDW bilayers. **b** Nonmagnetic and spin-polarized band structures, and spin density, of a seven-layer slab with *AL* stacking, where the top surface is terminated with a single CDW layer. The label in parentheses, *L*, indicates the stacking interface between the surface single layer and the bilayer beneath it. In the spin-polarized band structure, the majority and minority spin bands are represented by red and green lines, respectively. The shaded area (cyan, orange, and violet) associated with each band indicates the spectral weight of the central Ta $5d_{z^2}$ orbital in the top layer. **c** Projected density of states (PDOS) of the central Ta $5d_{z^2}$ orbital in the top layer of the slabs shown in (**a**) and (**b**), and the STM $dI/dV$ spectra[19].

beneath the surface forming a 2D Mott insulator in the third layer, its top-layer PDOS is nearly identical to that of the plain bilayer-terminated surface, with both displaying a peak-to-peak gap of around 0.26 eV (Fig. 5b). This similarity indicates that STM measurements would observe a large gap of 0.4 eV with peaks at ±0.2 eV on bilayer-terminated surfaces, irrespective of the presence of a single CDW layer in the third layer. The STM observations can then be accounted for as follows (Fig. 6a): When an *AL*-stacked bulk is cleaved, resulting in a single-layer step, one of the two terraces is terminated with a single layer. If this unstable single-layer-terminated terrace expands enough to overcome boundary effects, a spontaneous reconstruction of the CDWs is triggered, causing the single CDW layer in the first layer to relocate to the third layer. The resulting stabilized surface step exhibits bilayer termination on both terraces, each yielding a large-gap spectrum (refer to Supplementary Fig. 3 for the CDW reconstruction in two consecutive steps).

The proposed CDW reconstruction on the surface of 1T-TaS₂ has two notable characteristics. Firstly, it involves a CDW shift only in the second layer, while the other layers remain unaltered (Fig. 6a). Despite this localized change, the surface electronic structure undergoes a drastic transition from a Mott insulator to a band insulator. Secondly, the CDW shift takes place through the reconstruction of the existing SoD distortion, not via direct atomic displacement (Fig. 6b). This shift involves only modest atomic displacements around their undistorted lattice points, with a maximum lateral displacement of 0.4 Å. Nevertheless, these atomic motions collectively result in a 5.8 Å shift in the SoD position. This concerted atomic movement, without the addition or removal of atoms, highlights the uniqueness of the CDW surface reconstruction. It contrasts with conventional surface reconstructions observed in semiconductors and metal oxides, which typically

necessitate adding or removing atoms to fulfill local charge states and minimize broken bond counts[39].

## Discussion

The identification of CDW reconstruction in single-layer steps represents an important finding. It reveals that the majority of the 1T-TaS₂ surface, which exhibits a large gap, is not simply the plain bilayer-terminated surface but rather comprises two types of bilayer-terminated surfaces: the plain one and the one with a subsurface single CDW layer. This finding holds significant implications for surface observations, not only with STM but also with all surface-sensitive tools.

The observation of two bands with distinct $k_z$ dispersions around −0.2 eV in the ARPES measurements[8,34] can be attributed directly to the presence of two types of bilayer-terminated surfaces. Our findings indicate that the plain bilayer-terminated surface is the origin of the dispersive band, while the flat band arises from the bilayer surface with a subsurface single layer. Because of Coulomb repulsion, the Mott-insulating subsurface layer prevents electronic hopping between the surface bilayer and the subsurface bulk, causing the surface bilayer to be decoupled from the bulk and exhibit a flat dispersion along the $k_z$ direction. It is important to note that the flat band at −0.2 eV cannot be attributed to the Mott-insulating surface single layer because the corresponding STM $dI/dV$ spectrum shows a broad double-peak structure, as depicted in Fig. 3c.

Our findings also shed new light on surface observations made through time-resolved photoemission spectroscopy (tr-PES) and scanning thermoelectric microscopy (SThEM). The previous tr-PES measurements[11] observed an ultrashort-lived peak around +0.2 eV and broad spectral features in the gap region. A recent theoretical study

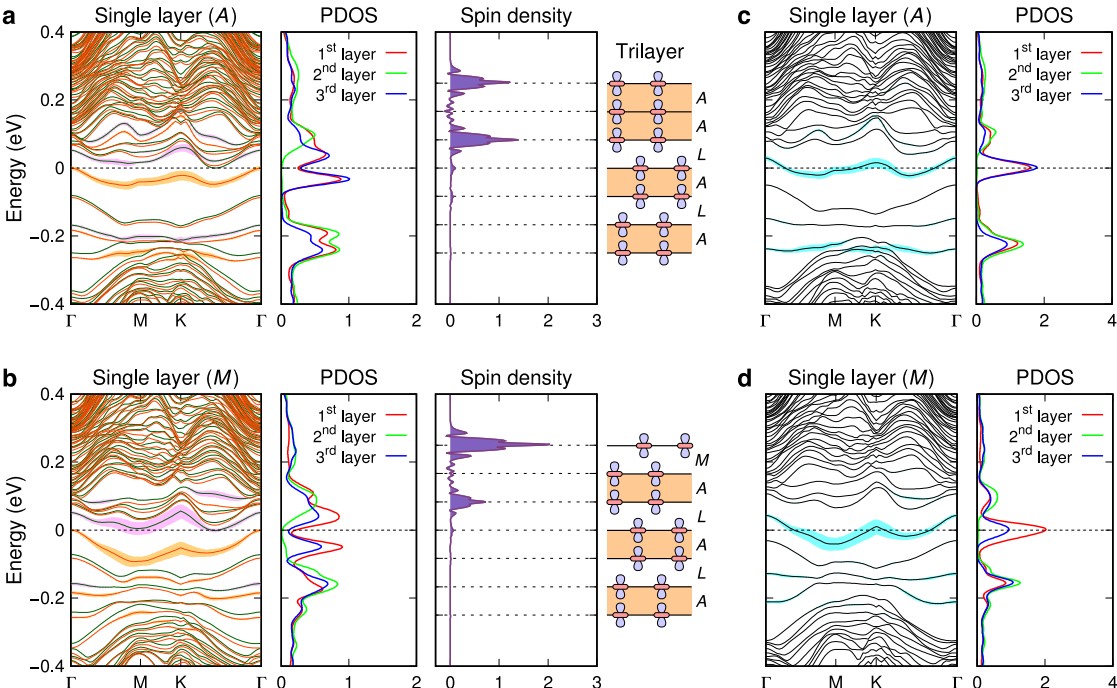

**Fig. 4 | Electronic structures of the "metallic" surfaces. a, b** Spin-polarized band structure, PDOS, and spin density of a seven-layer slab with *AL* stacking, where the top surface is terminated by a single layer with the *A* interface (**a**) and the *M* interface (**b**). **c, d** Nonmagnetic band structure and PDOS correspond to the same slab configurations as in (**a**, **b**), respectively. In the spin-polarized band structures, the majority and minority spin bands are represented by red and green lines, respectively. The shaded area (orange, violet, and cyan) associated with each band indicates the spectral weight of the central Ta $5d_{z^2}$ orbital in the top layer.

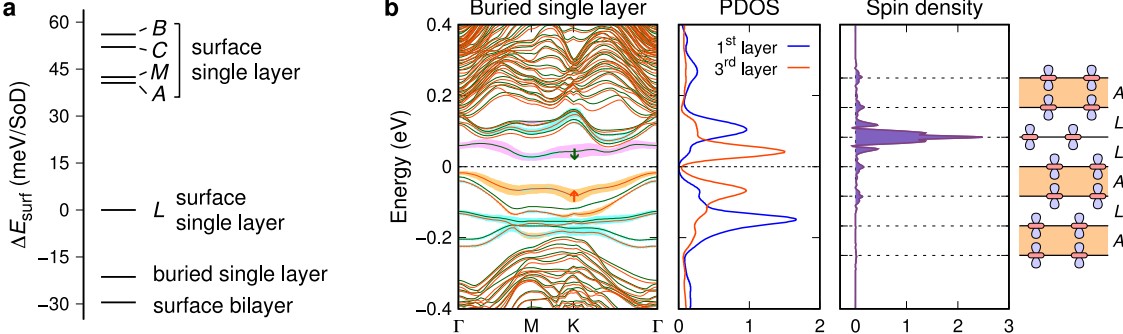

**Fig. 5 | Surface energies and a buried-single-layer surface. a** Surface energies of various surfaces [meV/($\sqrt{13} \times \sqrt{13}$)]. **b** Spin-polarized band structure, PDOS, and spin density of a seven-layer slab with a buried single layer. In the spin-polarized band structure, the majority and minority spin bands are represented by red and green lines, respectively. The shaded area in cyan (orange and violet) on the band structure represents the spectral weight of the central Ta $5d_{z^2}$ orbital in the first (third) layer.

utilizing dynamical mean-field theory simulations[40] attributed the sharp peak to the single-layer-terminated surface and the in-gap states to the plain bilayer-terminated surface, assuming that the former constituted a substantial portion of the surface. However, our results demonstrate that this assumption is incorrect, and we suggest that the peak at +0.2 eV originates from a bilayer surface that is decoupled from the bulk. The presence of peaks at ±0.2 eV in the bilayer surface corroborates this interpretation. Similarly, in the SThEM study[24], unexpected variations in thermopower were ascribed to the Mott-insulating single-layer-terminated surface, but our findings suggest that this interpretation needs to be reevaluated.

Lastly, we note that the 1T-TaS₂ surface presents a promising opportunity to study the trilayer Hubbard model. To understand the impact of interlayer coupling on the strongly correlated 2D system of cuprate high-temperature superconductors, the bilayer Hubbard model has been extensively studied, and the model at half-filling has been found to undergo a transition from a Mott insulator to a band insulator as interlayer coupling increases[41–43]. However, the trilayer Hubbard model, which more accurately represents the properties of multiply stacked materials in three dimensions, has received less attention. Our investigation of the 1T-TaS₂ surface has revealed that the "metallic" surfaces, which are, in fact Mott-insulating, correspond to trilayer Hubbard model systems in the strong interlayer coupling regime ($t_\perp \gg t_\parallel$). These findings indicate that the 1T-TaS₂ surface is a rare system for exploring the properties of the trilayer Hubbard model, prompting further research.

In conclusion, our investigation of the electronic properties of 1T-TaS₂ surfaces has revealed a new phenomenon: the surface reconstruction of CDWs in van der Waals layered materials. This finding clarifies several previously unresolved observations on the surface.

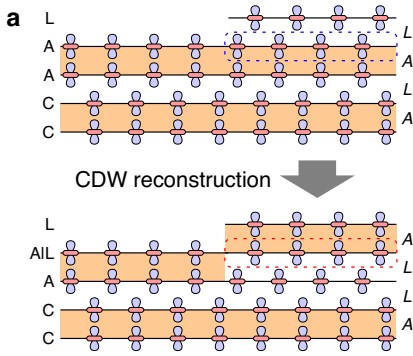

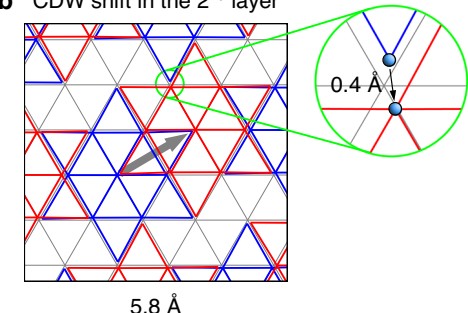

**Fig. 6 | CDW surface reconstruction. a** Surface reconstruction of the *L*-interface single-layer surface, located on the right side of a single-layer step, into a buried-single-layer surface. This process involves a CDW shift in the second layer, with no modifications made to the other layers. **b** CDW shift in the second layer. It induces a displacement of the SoD center from the "A" site (blue SoD) to the "L" site (red SoD), resulting in a shift of 5.8 Å. The magnified image of the region encircled in green illustrates the lateral displacement of a Ta atom by 0.4 Å. The gray triangular lattice denotes the undistorted Ta sublattice.

Our study underscores the necessity of considering CDW surface reconstruction when analyzing data from the surfaces of CDW-ordered layered materials, particularly in cases of strong interlayer coupling. Furthermore, our theoretical work has provided a detailed insight into strongly correlated phenomena in 1T-TaS$_2$. This highlights its potential as a model system for investigating the physics of correlated layered materials, especially the interplay between CDW, electron correlation, and interlayer coupling.

## Methods

### DFT calculations

We performed DFT + *U* calculations using the Vienna ab initio simulation package (VASP)[44,45]. These calculations employed the projector-augmented-wave method[46], the generalized gradient approximation[47], and the DFT + *U* scheme developed by Dudarev et al.[38]. To account for van der Waals interactions, we used the Tkatchenko–Scheffler approach[48]. Spin-orbit interactions were not incorporated, as previous calculations determined their effects to be insignificant[12,14,49]. The electronic wave functions were expanded using a plane-wave basis set, with cutoff energies of 323 and 259 eV for bulk and surface calculations, respectively. We carried out **k**-space integration using a $4 \times 4 \times 8$ mesh in the Brillouin zone of the $\sqrt{13} \times \sqrt{13} \times 1$ supercell. To optimize the lattice constants and atomic positions for each stacking configuration, we used the variable cell optimization method implemented in VASP. For *AL* stacking and *U* = 1.25 eV for Ta 5*d* orbitals, the calculated lattice constants were *a* = 3.36 Å and *c* = 5.77 Å, compared to experimental values of *a* = 3.37 Å and *c* = 5.90 Å[18]. The agreement between theory and experiment is quite good, especially given that the *c* parameter is mostly determined by van der Waals interactions. To model the surface, we employed a periodic slab geometry, where each slab comprised seven or eight TaS$_2$ layers, and the thickness of the vacuum region was 20 Å. We relaxed all atoms until all residual forces were less than 0.01 eV/Å.

## Data availability

The data used to generate the figures can be accessed in figshare (https://doi.org/10.6084/m9.figshare.23998236). Additional data are available from the corresponding authors upon reasonable request.

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

## Acknowledgements

This work was supported by the National Research Foundation of Korea (Grant No. 2018R1A2B6004044 (S.-H.L.), 2019K1A3A1A18116063 (S.-H.L.), 2017R1A5A1014862 (D.C.), 2020R1C1C1007895 (D.C.), and RS-2023-00251265 (D.C.)) and the Yonsei University Research Fund of 2019-22-0209 (D.C.).

## Author contributions

S.-H.L. and D.C. conceived the project, analyzed the results, and wrote the paper. S.-H.L. performed the calculation.

## Competing interests

The authors declare no competing interests.
