## [Peer Review File · Nature Communications]

REVIEWER COMMENTS

Reviewer #1 (Remarks to the Author):

The transition metal dichalcogenide 1T-TaS₂ is an intriguing van der Waals layered material with charge density wave (CDW) order and possible strong-correlation electronic states. The stacking order has recently been found to be closely related to the electronic state. This paper uses density functional theory calculation to study different stacking orders and corresponding electronic states, trying to get a picture consistent with the experimental results. They especially bring a scheme of CDW reconstruction and explain how this reconstruction transforms the surface state from a Mott insulator to a band insulator. However, I have several comments and questions as below:

1. The CDW reconstruction is not a unique phenomenon. From the author's picture in Fig. 6a, this CDW reconstruction corresponds to a domain wall aligned with the step edge. However, in the microscopic STM experiment, people never find hints of a domain wall aligned with the step edge. The author may suggest that this domain wall can be hidden underlying the top surface layer. In fact, occasionally, a domain wall can be observed below the top flat surface. The top flat surface and the domain wall in the underlying 2nd layer mean two different stackings on two sides of the domain wall. At the same time, they both show a large gap spectrum. This phenomenon, on the other hand, is completely in conflict with the author's picture in which the large gap is always related to a bilayer termination with AA stacked order.
2. The author's theory adheres to the previous point that the large energy gap comes from the bilayer band insulator, while the small gap comes from the L-interface single-layer surface. The AL stacking is calculated to be the state with the lowest energy. The surface energy calculation shows that the bilayer-terminated surface is the most stable one. Whether this AL stacking periodic structure exists is still under debate. For example, a three-layer AA-stacked step edge is shown in Ref. 23.
3. The DFT calculation tried to be consistent with very recent experimental results. The difference between the new calculations and previous DFT results should be explained. It seems that the DFT calculation in this system is a hard question. More deep thinking is required to address this issue.

With the above comments, I have to decline this paper for publication in Nature Communications.

Reviewer #2 (Remarks to the Author):

The manuscript reports on first-principles calculations of 1T-TaS₂ structures and is motivated by a discrepancy in the electronic structure reported for this material in the CDW phase where the differences arise when making comparisons between bulk calculations of the electronic structure of TaS₂ and surface sensitive measurements of TaS₂, for example by tunneling or ARPES. To explain the presence of the larger gap in the STM measurements reported in Ref. 23 the authors point to the possibility of a reconstruction of the TaS₂ top surface. To rationalize this the authors present first principles calculations of the electronic structure of different stacking configurations of few-layer TaS₂ structures where the top layer is either terminated with a single layer or appears as a sub-surface layer in between the bilayers that take on the ground state stacking.

There are two principal concerns I have that the authors should address to support the claims they make here. First the proposed reconstruction mechanism is discussed qualitatively in Figure 6. In the experiments conducted in Ref. 23 that the authors reference the sample is cleaved and measured at liquid nitrogen temperature (i.e on the order of few meV energy) while the calculated energy difference between the different stackings is several tens of meV. How do the authors explain this mismatch in energy scales? I don't think vibrational entropy provides an explanation for this difference.

Second - spin orbit interaction is not considered in this study. What role does it play in the electronic structure in particular with the different gaps predicted?

Third - In the context of correlating these calculated gaps with experiment it is important to note that the experiments being compared to are STM measurements. If indeed this sub-layer contributes to the lower gap one would expect the peak height in STM to be orders of magnitude compared to the principal band gap due to the larger exponential decay associated with the states in the sub surface layer with respect to the STM tip. I assume the states contributing are similar, i.e they are still Ta d_{z²} states? This does not appear to be the case of Ref. 23 where the peak heights appear to be similar. Can the authors explain this important discrepancy?

It is important that these three principal issues are *clearly* addressed to make this manuscript suitable for publication.

There are a couple of other minor comments that are mostly to do with formatting issues. For example I find referring to these structures as bilayers is misleading given that the calculations are done using few-layer structures. Similarly using "single layer" to label the different band structures in Figure 4 is also misleading since they are referring to the top surface layer in a few-layer structure.

Please find a new terminology for these figures. The title is highly misleading and makes this appear as a mechanism that applies to all vdW materials for which there is no evidence. Please rephrase.

Reviewer #3 (Remarks to the Author):

The authors present computational data that suggest a highly unusual surface relaxation mechanism for 1T-TaS₂ surfaces. The results are interesting, as they provide an explanation for various seemingly contradicting findings on the surface electronic properties of an intensively investigated material. Furthermore, the mechanism proposed by the authors might be relevant for further materials prone to charge density wave formation.

The present work is based on a previous study by the present authors (Ref. 15), but contains substantial new information. The present work (in contrast to Ref. 15) is based on DFT+U, with a U value determined essentially by comparison to experimental data for the bulk. This sets the study also apart from a series of previous theory works on 1T-TaS₂ surfaces, which were based on the DFT+U approach in Dudarev's approximation. The usage of DFT+U to describe electron correlation effects is certainly a limitation of the present work. However, this limitation is hard to overcome for systems such as studied here and the authors present a clear rationale for their choice of method. Also, sufficient detail is provided to allow for reproducing the calculations.

All in all, the paper is enjoyable to read and provides an interesting

and original attempt to explain a number of puzzling results for an important model system. I am therefore in favor of publication.

However, I am not really sure if the title is appropriate: Surface reconstruction typically refers to the change of the surface unit cell compared to the truncated bulk, due to lateral forces. The phenomenon discussed by the authors essentially concerns the vertical interaction between the layers and is perhaps better called surface relaxation.

Reviewer #1 (Remarks to the Author):

The transition metal dichalcogenide 1T-TaS₂ is an intriguing van der Waals layered material with charge density wave (CDW) order and possible strong-correlation electronic states. The stacking order has recently been found to be closely related to the electronic state. This paper uses density functional theory calculation to study different stacking orders and corresponding electronic states, trying to get a picture consistent with the experimental results. They especially bring a scheme of CDW reconstruction and explain how this reconstruction transforms the surface state from a Mott insulator to a band insulator. However, I have several comments and questions as below:

1. The CDW reconstruction is not a unique phenomenon. From the author's picture in Fig. 6a, this CDW reconstruction corresponds to a domain wall aligned with the step edge. However, in the microscopic STM experiment, people never find hints of a domain wall aligned with the step edge.

We appreciate the reviewer's comment. Indeed, the possibility of domain boundaries forming beneath the step edge had not been previously considered, neither by us nor by anyone else. There was no a priori justification for domain boundaries to emerge in that location. However, by focusing our attention on flat terraces and examining their stability with respect to CDW stacking, the computed results and their comparison with experimental data unambiguously demonstrate that a domain boundary ought to exist immediately beneath the step edge. This shift in focus was precisely what we intended when we introduced the term "CDW surface reconstruction."

The author may suggest that this domain wall can be hidden underlying the top surface layer. In fact, occasionally, a domain wall can be observed below the top flat surface. The top flat surface and the domain wall in the underlying 2nd layer mean two different stackings on two sides of the domain wall. At the same time, they both show a large gap spectrum. This phenomenon, on the other hand, is completely in conflict with the author's picture in which the large gap is always related to a bilayer termination with AA stacked order.

In response to the reviewer's concern, we would like to suggest that the domain boundary is more likely to reside in the third layer beneath a surface CDW bilayer, as opposed to being in the second layer. Domain boundaries in the third layer can disrupt the electronic structure of the second layer, as well as the interlayer hybridization between the first and second layers. As a result, these domain boundaries would leave noticeable traces in the PDOS of the first layer, which otherwise exhibits a large gap spectrum. Considering the substantial plausibility of this alternative explanation, we respectfully propose that the instance brought up by the reviewer might not serve as a counterexample to our theory.

2. The author's theory adheres to the previous point that the large energy gap comes from the bilayer band insulator, while the small gap comes from the *L*-interface single-layer surface. The *AL* stacking is calculated to be the state with the lowest energy. The surface energy calculation

shows that the bilayer-terminated surface is the most stable one. Whether this *AL* stacking periodic structure exists is still under debate. For example, a three-layer AA-stacked step edge is shown in Ref. 23.

We would like to point out that the definitive confirmation or refutation of bilayer CDW stacking in 1T-TaS₂ cannot be achieved solely through STM, as it is limited to probing the material's surface. Conversely, numerous experiments that delve into the material's internal periodic structure have furnished evidence supporting bilayer CDW stacking. Such experiments include low-energy electron diffraction (Ref. 16), X-ray diffraction (Refs. 17, 18, 30), nuclear magnetic/quadrupole resonance (Refs. 31, 32), and transmission electron microscopy (Ref. 33). Additionally, DFT calculations have demonstrated that the ARPES spectrum is significantly influenced by CDW stacking, and only *AL* stacking aligns with the experimental data (Refs. 13, 14, 15). Given this array of evidence, we contend that the question of bilayer *AL* stacking is largely resolved, and the remaining task is to make sense of observations that seem to contradict bilayer CDW stacking, which has been the motivation for our current research.

The hypothesis of three-layer CDW stacking was proposed based on the observation of two consecutive AA-type step edges, as shown in Fig. 7 of Ref. 23. It is worth noting that (i) Fig. 7 is located in the appendix section rather than the main results section, (ii) it was included during the final revision of the paper (refer to earlier versions of the paper at the arxiv site, <http://128.84.21.203/abs/2105.08663>), and (iii) the quality of the STM images in Fig. 7 is suboptimal, displaying multiple replicas of the step edge that are not shown in other figures. Consequently, it can be inferred that instances of two consecutive AA-type step edges are exceedingly rare, indicating that at least one of the three terraces is composed of energetically unstable surfaces involving the *B*, *C*, or *M* stacking interfaces in the subsurface layers. (Indeed, the situation depicted in Fig. 7 can be explained by positing that the top terrace is a bilayer-terminated surface with a single CDW layer in the third layer, accompanied by an unstable interface between the surface bilayer and the single layer.) We respectfully suggest that this infrequent occurrence involving unstable terraces may not offer ample evidence to dispute the bilayer CDW stacking in bulk 1T-TaS₂.

Finally, we would like to underscore that our proposed theory of CDW surface reconstruction, based on bilayer CDW stacking, comprehensively accounts for the primary data presented in Ref. 23, including all the figures in the main results section, as well as Fig. 8 in the appendix section.

3. The DFT calculation tried to be consistent with very recent experimental results. The difference between the new calculations and previous DFT results should be explained. It seems that the DFT calculation in this system is a hard question. More deep thinking is required to address this issue.

We are grateful for the reviewer's comment. In the past, DFT calculations failed to explain the gap opening in bulk 1T-TaS₂ and to reproduce the ARPES spectrum on the surface, until

comprehensive investigations of the CDW stacking order were presented in Ref. 14 and 15. This prolonged period of failure led to the prevalent notion that DFT calculations might not accurately portray 1T-TaS₂. Nonetheless, as Ref. 15 demonstrated, a systematic examination of the CDW stacking order and its corresponding energetic stability could shed light on various experimental findings related to 1T-TaS₂. In accordance with this perspective, our study has expanded the analysis of the CDW stacking order and energetic stability to the surface of 1T-TaS₂, and intriguingly, we were able to offer explanations for some previously puzzling observations by introducing the concept of CDW surface reconstruction.

There are two main differences between the present DFT results and those of earlier studies. First, most previous research did not conduct a systematic analysis of the energetic stability of bulk and surface 1T-TaS₂ as a function of CDW stacking order. Instead, these studies focused on the electronic structures of a few select cases, such as *A* stacking, *L* stacking, or *AL* stacking, without evaluating their relative stability. Second, some earlier research utilized large *U* values and concluded that 1T-TaS₂ is a Mott insulator based solely on the gap opening induced by layer-by-layer antiferromagnetic order, without examining its energetic stability or making thorough comparisons with experimental observations. Given these differences, it is reasonable to assert that our DFT results are in agreement with previous research for the specific cases and large *U* cases. In our manuscript, we have not explicitly focused on the limitations of past studies; instead, we have provided an up-to-date understanding of 1T-TaS₂, while citing noteworthy theoretical works where appropriate and addressing the issue of proper *U* values at the beginning of the Results section. We sincerely hope that the reviewer understands and values our approach.

With the above comments, I have to decline this paper for publication in Nature Communications.

We appreciate the reviewer's comments and sincerely hope that this response has resolved their concerns to a considerable extent.

After submitting our manuscript, we identified an error in our calculation of the buried-single-layer surface. The rectified calculation indicates that the energy gain resulting from the CDW surface reconstruction has decreased to two-thirds of the original value ($32 \rightarrow 21$ meV/ $(\sqrt{13} \times \sqrt{13})$), making the reconstructed, buried-single-layer surface ~ 10 meV/ $(\sqrt{13} \times \sqrt{13})$ less stable than the plain bilayer-terminated surface (refer to Fig. 5a). Notably, this updated value is more consistent with the results that *AL* stacking is more stable than *L* stacking for bulk 1T-TaS₂. Despite the reduction in energy gain, the revised value of 21 meV/ $(\sqrt{13} \times \sqrt{13})$ remains significant enough that no further changes to the manuscript are necessary, apart from the inclusion of the updated value. We respectfully request the reviewer's understanding regarding this update.

Reviewer #2 (Remarks to the Author):

The manuscript reports on first-principles calculations of 1T-TaS₂ structures and is motivated by a discrepancy in the electronic structure reported for this material in the CDW phase where the differences arise when making comparisons between bulk calculations of the electronic structure of TaS₂ and surface sensitive measurements of TaS₂, for example by tunneling or ARPES. To explain the presence of the larger gap in the STM measurements reported in Ref. 23 the authors point to the possibility of a reconstruction of the TaS₂ top surface. To rationalize this, the authors present first principles calculations of the electronic structure of different stacking configurations of few-layer TaS₂ structures where the top layer is either terminated with a single layer or appears as a sub-surface layer in between the bilayers that take on the ground state stacking.

There are two principal concerns I have that the authors should address to support the claims they make here. First the proposed reconstruction mechanism is discussed qualitatively in Figure 6. In the experiments conducted in Ref. 23 that the authors reference the sample is cleaved and measured at liquid nitrogen temperature (i.e., on the order of few meV energy) while the calculated energy difference between the different stackings is several tens of meV. How do the authors explain this mismatch in energy scales? I don't think vibrational entropy provides an explanation for this difference.

We appreciate the valuable comment provided by the reviewer. We would like to highlight that the surface energy difference between different stackings is several tens of meV per ($\sqrt{13} \times \sqrt{13}$). As the area of $\sqrt{13} \times \sqrt{13}$ contains 13 Ta atoms and 26 S atoms per layer, the energy difference corresponds to 1--2 meV per atom in the top layer. Therefore, the thermal energy at liquid nitrogen temperature is sufficient for stabilizing the surface CDW stacking. In addition, there is another important source of energy: the cleaving process. The absolute value of the surface energy is calculated to be around 2.3 eV / ($\sqrt{13} \times \sqrt{13}$). Thus, it is expected that the cleaving process will impart surface atoms with kinetic energy of several tens of meV per atom (i.e., an order of eV / ($\sqrt{13} \times \sqrt{13}$)), which is significant enough for them to reach lower-energy CDW stacking configurations or even unstable higher-energy stacking configurations for small domains. We have included this discussion in the manuscript.

Second - spin orbit interaction is not considered in this study. What role does it play in the electronic structure in particular with the different gaps predicted?

For 1T-TaS₂, previous DFT calculations have shown that the effects of spin-orbit interactions (SOI) on the band gap are negligible, with the modification of an order of 0.01 eV. Specifically, the band gap of the band-insulating bulk 1T-TaS₂ in *AL* stacking was changed by only 0.014 eV by the inclusion of SOI in GGA calculations (refer to Sec. II.3 and Fig. 2(d) of Ref. 14). The Mott gap of a single-layer 1T-TaS₂ of 0.22 eV in GGA + *U* calculations was apparently unchanged by SOI (refer to Fig. 3(a) of Ref. 49). We have included a statement in the Method section of the manuscript, to explain the rationale for not incorporating SOI in our calculations,

citing relevant references.

Third - In the context of correlating these calculated gaps with experiment it is important to note that the experiments being compared to are STM measurements. If indeed this sub-layer contributes to the lower gap one would expect the peak height in STM to be orders of magnitude compared to the principal band gap due to the larger exponential decay associated with the states in the sub surface layer with respect to the STM tip. I assume the states contributing are similar, i.e. they are still $Ta dz^2$ states? This does not appear to be the case of Ref. 23 where the peak heights appear to be similar. Can the authors explain this important discrepancy?

We wish to provide clarity that the subsurface layer does *not* contribute to the lower gap spectrum in STM measurements. As correctly pointed out by the reviewer, the contribution from the subsurface layer would be orders of magnitude smaller than that of the surface layer. Although not explicitly stated, the reviewer appears to have compared the PDOS of Fig. 5b in our paper with the dI/dV spectra in Fig. 5(b) of Ref. 23, which display similar features. If this is indeed the case, we would like to emphasize that such a comparison would be inappropriate. Instead, the red line in the dI/dV spectra in Fig. 5(b) of Ref. 23 should be compared with the top-layer PDOS of the single-layer-terminated surface with the B or M interface ($T_s = \pm a + c$) (Fig. 4b in our study), rather than the third-layer PDOS of the buried-single-layer surface. In response to the reviewer's comment, we have revised the text describing the PDOS of Fig. 5(b) to prevent any potential confusion.

It is important that these three principal issues are **clearly** addressed to make this manuscript suitable for publication.

We sincerely hope that our response has been satisfactory to the reviewer.

There are a couple of other minor comments that are mostly to do with formatting issues. For example, I find referring to these structures as bilayers is misleading given that the calculations are done using few-layer structures. Similar using "single layer" to label the different band structures in Figure 4 is also misleading since they are referring to the top surface layer in a few-layer structure. Please find a new terminology for these figures.

We appreciate the comment provided by the reviewer. In response, we have made modifications in Fig. 3 by changing the label 'bilayer' to 'bilayer surface' and the label 'single layer (L)' to 'single layer (L) surface', along with elaborating the corresponding captions. In Fig. 4, due to the limited space available, we have retained the short labels, but revised the caption to clarify that the data were obtained from seven-layer slabs with a single-layer-terminated top surface, rather than a single layer.

The title is highly misleading and makes this appear as a mechanism that applies to all vdW materials for which there is no evidence. Please rephrase.

In acknowledgement of the reviewer's observation, we have revised the title of our research from "... in van der Waals materials" to "... in a van der Waals material." We would like to emphasize that this change should not be interpreted as suggesting that the phenomenon of CDW surface reconstruction is limited solely to 1T-TaS₂. A wide variety of critical layered materials, including transition metal dichalcogenides (AB₂, where A represents Nb, Ta, or Ti and B signifies S or Se) and kagome superconductors (AV₃Sb₅, where A denotes Cs, Rb, or K), exhibit pronounced three-dimensional CDW order accompanied by substantial interlayer coupling. These layered materials, depending on the cleavage plane, could undergo significant surface reconstruction of CDWs, which in turn could affect profound modifications in the surface electronic and chemical properties.

We appreciate the reviewer's feedback and thank them for their contribution to the improvement of our manuscript.

After submitting our manuscript, we identified an error in our calculation of the buried-single-layer surface. The rectified calculation indicates that the energy gain resulting from the CDW surface reconstruction has decreased to two-thirds of the original value ($32 \rightarrow 21 \text{ meV}/(\sqrt{13}\times\sqrt{13})$), making the reconstructed, buried-single-layer surface $\sim 10 \text{ meV}/(\sqrt{13}\times\sqrt{13})$ less stable than the plain bilayer-terminated surface (refer to Fig. 5a). Notably, this updated value is more consistent with the results that *AL* stacking is more stable than *L* stacking for bulk 1T-TaS₂. Despite the reduction in energy gain, the revised value of $21 \text{ meV}/(\sqrt{13}\times\sqrt{13})$ remains significant enough that no further changes to the manuscript are necessary, apart from the inclusion of the updated value. We respectfully request the reviewer's understanding regarding this update.

Reviewer #3 (Remarks to the Author):

The authors present computational data that suggest a highly unusual surface relaxation mechanism for 1T-TaS₂ surfaces. The results are interesting, as they provide an explanation for various seemingly contradicting findings on the surface electronic properties of an intensively investigated material. Furthermore, the mechanism proposed by the authors might be relevant for further materials prone to charge density wave formation.

The present work is based on a previous study by the present authors (Ref. 15), but contains substantial new information. The present work (in contrast to Ref. 15) is based on DFT+U, with a U value determined essentially by comparison to experimental data for the bulk. This sets the study also apart from a series of previous theory works on 1T-TaS₂ surfaces, which were based on the DFT+U approach in Dudarev's approximation. The usage of DFT+U to describe electron correlation effects is certainly a limitation of the present work. However, this limitation is hard to overcome for systems such as studied here and the authors present a clear rationale for their choice of method. Also, sufficient detail is provided to allow for reproducing the calculations.

All in all, the paper is enjoyable to read and provides an interesting and original attempt to explain a number of puzzling results for an important model system. I am therefore in favor of publication.

We would like to express our sincere gratitude to the reviewer for their complimentary comments and their recommendation for publication.

However, I am not really sure if the title is appropriate: Surface reconstruction typically refers to the change of the surface unit cell compared to the truncated bulk, due to lateral forces. The phenomenon discussed by the authors essentially concerns the vertical interaction between the layers and is perhaps better called surface relaxation.

We appreciate the reviewer's comment regarding the terminology, to which we had also devoted considerable thought. Despite the fact that the surface unit cell size remains constant during the surface modification, thus implying surface relaxation, it is also true that bond breaking and formation occur on the surface, indicating surface reconstruction. To elucidate this apparent ambiguity, we wish to underscore that the CDW-ordered bulk 1T-TaS₂ already represents a "reconstructed" state with a periodicity of $\sqrt{13}\times\sqrt{13}$, deviating from the ideal bulk crystalline structure. On the single-layer-terminated surface, the $\sqrt{13}\times\sqrt{13}$ reconstruction in the second layer undergoes a reconstruction of the existing reconstruction, leading to a lateral shift in the center of the reconstruction by 5.8 Å. Thus, it is more appropriate to refer to this alteration of the $\sqrt{13}\times\sqrt{13}$ reconstruction on the surface as surface reconstruction rather than surface relaxation. We hope that our response adequately addresses the reviewer's concern.

We appreciate the reviewer's feedback and thank them for their contribution to the

improvement of our manuscript.

After submitting our manuscript, we identified an error in our calculation of the buried-single-layer surface. The rectified calculation indicates that the energy gain resulting from the CDW surface reconstruction has decreased to two-thirds of the original value ($32 \rightarrow 21$ meV/ $(\sqrt{13} \times \sqrt{13})$), making the reconstructed, buried-single-layer surface ~ 10 meV/ $(\sqrt{13} \times \sqrt{13})$ less stable than the plain bilayer-terminated surface (refer to Fig. 5a). Notably, this updated value is more consistent with the results that AL stacking is more stable than L stacking for bulk 1T-TaS₂. Despite the reduction in energy gain, the revised value of 21 meV/ $(\sqrt{13} \times \sqrt{13})$ remains significant enough that no further changes to the manuscript are necessary, apart from the inclusion of the updated value. We respectfully request the reviewer's understanding regarding this update.

REVIEWER COMMENTS

Reviewer #1 (Remarks to the Author):

First, I have a comment about the concept of “surface reconstruction”. The CCDW phase appeared below the NCCDW-CCDW transition temperature (170 K). The cleaving process in experiments can happen either at liquid nitrogen (LN2) temperature of 77 K or at room temperature. For the LN2 cleaving, the CCDW phase exists before cleaving. If the theoretical scheme in this manuscript is followed, the CCDW phase on the cleaved surface may change after cleaving, called the surface reconstruction by the authors. On the other hand, for the room temperature cleaving, the boundary conditions of step edges appear first. As the temperature decreases and the phase transition temperature is reached, the CCDW phase begins to appear, whose structure in the bulk and on the surface follows the principle of minimum energy. The buried-single-layer surface can similarly appear around the step edge with the same theoretical scheme. I am not sure whether it can still be called a “surface reconstruction”.

The authors try to give a comprehensive scheme to explain existing experiments. However, I am still not convinced by this scheme. Following I present two of the main concerns.

1) The authors mention that there are different experiments including “low-energy electron diffraction (Ref. 16), X-ray diffraction (Refs. 17, 18, 30), nuclear magnetic/quadrupole resonance (Refs. 31, 32), and transmission electron microscopy (Ref. 33)” giving evidence of bilayer CDW stackings. From my understanding, they are all INDIRECT evidence of the AL bilayer, although I believe the dimerization may exist in some forms. A very recent paper “Atomic visualization of the 3D charge density wave stacking in 1T-TaS₂ by cryogenic transmission electron microscopy” (<https://doi.org/10.1021/acs.nanolett.3c00556>) gives relatively direct evidence about the CDW stacking picture: “Taken together, the realistic interlayer stacking of SODs in 1T-TaS₂ should be dominated by the TA configuration, while in some layers, partial SODs shift from the aligned centers to site B or C, forming local TB- or TC-stacked interfaces. The upper layers of the interface continue to appear in the configuration of TA stacking, but B or C stacking domains are formed relative to the reference layer in the z-direction (Figure 3f). Considering the in-plane configurations of an interface, misaligned SODs may be encountered, forming the A-, B- and C-stacked domains separated by domain walls (Figure 3g).” The A-stacking is the dominant order in their 10 nm thick sample. The AL bilayer is not observed in this cryogenic TEM experiment. It remains a matter of discussion what the interlayer dimerization is, rather than the authors' belief that the AL stacking has been largely resolved.

2) About the domain wall issue in the previous report, here I give four different situations in the following figure (see the attached file) . The authors have agreed that a domain wall should appear below the step edge in their scheme. I give two cases in Fig. 1(a) and Fig. 1(b). The grey arrow labels how the surface reconstruction happens from the top diagram to the bottom diagram. The domain wall position is labeled by the red rectangle, appearing in the top layer and the 2nd layer, respectively. Please note that this is not a spontaneously appeared domain wall in the bulk sample, but a new domain wall compatible with the “surface reconstruction” picture. For this new domain wall: (a) Extra energy should be provided to have this new domain wall. (b) This new domain wall should be aligned with the step edge. In Fig. 1(c), a spontaneously appeared domain wall exists in the top layer (labeled by a yellow rectangle), which is a common phenomenon in STM experiments. Both sides of the domain wall usually show a large-gap spectrum. Compatible with the “surface reconstruction”, a new domain wall has to appear right below the natural domain wall in the top layer. Similarly, for this new domain wall: (a) Extra energy should be provided to have this new domain wall. (b) This new domain wall should be aligned with the natural domain wall in the top layer. On the right side of this “double domain wall”, four consecutive AA stackings exist. Fig. 1(d) describes the fourth case, where a natural domain wall exists below the top layer. The authors claim that the domain wall below the top layer should be in the 3rd layer instead of the 2nd layer. Even with this situation, there are two possible reconstructed structures in Fig. 1(d) (top and bottom diagrams). In the top diagram, there are three AA stackings on the right side of the domain wall, which is more like a metallic surface spectrum with the theoretical scheme. In the bottom picture, there are 6 consecutive AA stackings.

From these four different cases, it is hard for us to accept the “surface reconstruction” scheme. (1) It is not easy to make this scheme compatible with natural domain walls in the bulk sample. (2) In the theoretical calculation, the authors do not consider the extra energy required for the new domain wall, which is not a negligible quantity.

(3) There are never any hints about aligned domain walls in previous literature (either aligned with the step edge or aligned with the existing natural domain wall).

For the domain wall beneath the top layer, it is more reasonable to attribute it to the 2nd layer instead of the 3rd layer. Like in Ref. 8: “Apart from the well-defined domain walls in the top layer, domain walls in the layer underneath are clearly resolved as networks of random filamentary features.”

In summary, the scheme of surface reconstruction is not convincing enough and requires further investigations by the authors.

Reviewer #2 (Remarks to the Author):

I thank the authors for responding to my questions, all of which have been addressed. I recommend this article be published in Nature Communications.

Reviewer #3 (Remarks to the Author):

To my mind, the authors have satisfactorily responded to the referees comments and appropriately modified the manuscript. The paper is a very interesting and original study and should be published as is.

Reviewer #1 (Remarks to the Author):

First, I have a comment about the concept of “surface reconstruction”. The CCDW phase appeared below the NCCDW-CCDW transition temperature (170 K). The cleaving process in experiments can happen either at liquid nitrogen (LN2) temperature of 77 K or at room temperature. For the LN2 cleaving, the CCDW phase exists before cleaving. If the theoretical scheme in this manuscript is followed, the CCDW phase on the cleaved surface may change after cleaving, called the surface reconstruction by the authors. On the other hand, for the room temperature cleaving, the boundary conditions of step edges appear first. As the temperature decreases and the phase transition temperature is reached, the CCDW phase begins to appear, whose structure in the bulk and on the surface follows the principle of minimum energy. The buried-single-layer surface can similarly appear around the step edge with the same theoretical scheme. I am not sure whether it can still be called a “surface reconstruction”.

We sincerely appreciate the reviewer's thorough evaluation of our manuscript. In response to the first issue raised, we would like to highlight that surface reconstruction refers to the spontaneous reconfiguration of atomic structures on the surface of a material, aimed at reducing the surface (free) energy. In essence, surface reconstruction represents a direct manifestation of the "principle of minimum energy" on the surface. Consequently, the formation of a buried-single-layer surface during the cooling process can also be considered as a manifestation of surface reconstruction. This is because the buried-single-layer surface is energetically preferred over the surface with a single CDW layer directly on it.

The authors try to give a comprehensive scheme to explain existing experiments. However, I am still not convinced by this scheme. Following I present two of the main concerns.

1) The authors mention that there are different experiments including “low-energy electron diffraction (Ref. 16), X-ray diffraction (Refs. 17, 18, 30), nuclear magnetic/quadrupole resonance (Refs. 31, 32), and transmission electron microscopy (Ref. 33)” giving evidence of bilayer CDW stackings. From my understanding, they are all INDIRECT evidence of the AL bilayer, although I believe the dimerization may exist in some forms. A very recent paper “Atomic visualization of the 3D charge density wave stacking in 1T-TaS₂ by cryogenic transmission electron microscopy” (<https://doi.org/10.1021/acs.nanolett.3c00556>) gives relatively direct evidence about the CDW stacking picture: “Taken together, the realistic interlayer stacking of SODs in 1T-TaS₂ should be dominated by the TA configuration, while in some layers, partial SODs shift from the aligned centers to site B or C, forming local TB- or TC-stacked interfaces. The upper layers of the interface continue to appear in the configuration of TA stacking, but B or C stacking domains are formed relative to the reference layer in the z-direction (Figure 3f). Considering the in-plane configurations of an interface, misaligned SODs may be encountered, forming the A-, B- and C-stacked domains separated by domain walls (Figure 3g).” The A-stacking is the dominant order in their 10 nm thick sample. The AL bilayer is not observed in this cryogenic TEM experiment. It remains a matter of discussion what the interlayer dimerization is, rather than the authors' belief that the AL stacking has been largely resolved.

While the reviewer regards the recent TEM study as providing direct experimental evidence, it is important to note that the authors of the paper have not presented any direct TEM images. Instead, they have presented images that have undergone sophisticated aberration correction and Fourier bandpass filtering. Consequently, their images are highly susceptible to potential artifacts.

The TEM study asserts that the surface predominantly exhibits *A* stacking. However, this conclusion not only contradicts several previous experimental findings (Ref. 16-18, 30-33) but also significantly differs from the current theoretical understanding. Firstly, DFT studies have indicated that *A* stacking is significantly less stable than *AL* stacking due to the increased layer spacing in *A* stacking, which reduces the van der Waals energy gain (Ref. 15). The reasoning behind this increase in layer spacing in *A* stacking is quite plausible: under SoD distortion, where Ta atoms are drawn to the SoD center, the S atoms near the SoD center protrude outward. This results in interlayer S-S repulsion in the on-top SoD stacking configuration, leading to an increase in interlayer spacing. Secondly, the ARPES simulation for *A* stacking greatly deviates from the observed ARPES data (Ref. 13 & 15), whereas the simulation for *AL* stacking aligns exceptionally well with the experimental data (Ref. 14 & 15).

We raise doubts regarding the applicability of transmission electron microscopy (TEM) in accurately discerning the stacking of CDWs in a sample with a thickness of approximately 10 nm, corresponding to roughly 17 layers. Specifically, in the case of *AL* stacking, where the "*L*" interface comprises three symmetry-equivalent interfaces due to rotational symmetry, CDW bilayers form with interfaces randomly selected from these equivalent options, introducing stacking pattern randomness. Consequently, in a TEM image obtained through approximately 17 layers, the peaks associated with CDW stacking are prone to blurring and susceptible to elimination through Fourier bandpass filtering. This speculation cautiously offers a possible explanation for the identification of trivial *A* stacking in the TEM study.

2) About the domain wall issue in the previous report, here I give four different situations in the following figure. The authors have agreed that a domain wall should appear below the step edge in their scheme. I give two cases in Fig. 1(a) and Fig. 1(b). The grey arrow labels how the surface reconstruction happens from the top diagram to the bottom diagram. The domain wall position is labeled by the red rectangle, appearing in the top layer and the 2nd layer, respectively. Please note that this is not a spontaneously appeared domain wall in the bulk sample, but a new domain wall compatible with the "surface reconstruction" picture. For this new domain wall: (a) Extra energy should be provided to have this new domain wall. (b) This new domain wall should be aligned with the step edge. In Fig. 1(c), a spontaneously appeared domain wall exists in the top layer (labeled by a yellow rectangle), which is a common phenomenon in STM experiments. Both sides of the domain wall usually show a large-gap spectrum. Compatible with the "surface reconstruction", a new domain wall has to appear right below the natural domain wall in the top layer. Similarly, for this new domain wall: (a) Extra energy should be provided to have this new domain wall. (b) This new domain wall should be aligned with the natural domain wall in the top layer. On the right side of this "double domain

wall”, four consecutive AA stackings exist. Fig. 1(d) describes the fourth case, where a natural domain wall exists below the top layer. The authors claim that the domain wall below the top layer should be in the 3rd layer instead of the 2nd layer. Even with this situation, there are two possible reconstructed structures in Fig. 1(d) (top and bottom diagrams). In the top diagram, there are three AA stackings on the right side of the domain wall, which is more like a metallic surface spectrum with the theoretical scheme. In the bottom picture, there are 6 consecutive AA stackings.

From these four different cases, it is hard for us to accept the “surface reconstruction” scheme.

- (1) It is not easy to make this scheme compatible with natural domain walls in the bulk sample.
- (2) In the theoretical calculation, the authors do not consider the extra energy required for the new domain wall, which is not a negligible quantity.
- (3) There are never any hints about aligned domain walls in previous literature (either aligned with the step edge or aligned with the existing natural domain wall).

For the domain wall beneath the top layer, it is more reasonable to attribute it to the 2nd layer instead of the 3rd layer. Like in Ref. 8: “Apart from the well-defined domain walls in the top layer, domain walls in the layer underneath are clearly resolved as networks of random filamentary features.”

Let us begin by addressing Point (2). The energy required for the formation of domain boundaries during CDW surface reconstruction is directly proportional to the circumference of the domain subject to the CDW reconstruction. When the domain's area exceeds a certain threshold, the energy gained through reconstruction surpasses the energy cost associated with domain boundaries. Hence, we asserted in our manuscript that spontaneous CDW reconstruction would take place in sufficiently large domains featuring unstable CDW stacking on the surface. This is consistent with experimental findings, where energetically unstable single-CDW-layer surfaces are invariably observed in small domains. To trigger CDW reconstruction within the terrace region, extra energy may be required, possibly provided through thermal energy or cleaving processes.

Regarding Point (3), it's crucial to note that the edges of the upper layer at single-layer steps display edge states, which produce intricate energy spectra contingent on the precise atomic structures of the edge. This could significantly disrupt the traces of domain boundaries in the second layer. Fig. 2(c) of Ref. 23 presents an STS image across the step edge, which is presumed to possess a domain boundary beneath it. Notably, there are markedly disturbed

dI/dV spectra extending approximately 30 Å across the step edge. These dI/dV spectra exceed the capabilities of quantitative theoretical analysis due to the numerous possibilities of edge termination and the effect of convolution with the step height of 6 Å. Consequently, the lack of any indications of domain boundaries correlating with step edges in prior studies should not be interpreted as evidence disputing the possibility of CDW surface reconstruction.

The quotation from Ref. 8 suggests that the reviewer intends to utilize our CDW surface reconstruction theory to interpret the STM results presented in Ref. 8. However, we caution that our theory may not fully encompass the intricacies of the STM findings in Ref. 8. This limitation arises primarily due to the artificially generated surfaces in Ref. 8 through voltage pulses applied to the STM tip. As a result, the surface exhibits numerous unstable CDW stackings and domain boundaries in both surface and subsurface layers, which are not typically encountered in samples cooled slowly or cleaved at cryogenic temperatures. The introduced complexity, deviating significantly from the ground-state phase, lies beyond the scope of our work.

In response to Point (1), concerning the cases depicted in Fig. 1(c) and Fig. 1(d), let us address each case separately. Starting with Fig. 1(c), the reviewer expressed difficulty in comprehending the subsurface CDW stacking structure in the presence of a domain boundary on the top surface layer. We propose an alternative scenario that offers favorable energy considerations. Consider a boundary between two domains, where the SoD centers of the top four layers reside at A-A-L-L sites from the 4th layer to the 1st layer on the left terrace, while they are situated at A-A-H-H sites on the right terrace. This domain boundary is energetically favorable (except for the formation energy of the domain boundary itself) as both terraces exhibit stable *AL* stacking (note that *H* is a symmetry-equivalent interface with *L*). It is noteworthy that this model does not involve any domain boundaries in the third layer and below.

Turning to Fig. 1(d), the reviewer raised concerns about the implications of our theory when single CDW layers exist in the third and/or fourth layers. Central to our theory is the assertion that a single CDW layer on the surface, specifically the first layer, is energetically highly unfavorable and should be displaced to the third layer, leaving a surface bilayer. The presence of single layers in the third or fourth layers is not as unstable as in the first layer, as evidenced by the stability of the buried-single-layer surface. Consequently, our theory does not require a CDW stacking shift in the case depicted in Fig. 1(d).

In summary, the scheme of surface reconstruction is not convincing enough and requires further investigations by the authors.

We appreciate the reviewer so much for taking time to consider our theory seriously. We hope that our responses to your queries are reasonably satisfactory.

Reviewer #2 (Remarks to the Author):

I thank the authors for responding to my questions, all of which have been addressed. I recommend this article be published in Nature Communications.

We thank the reviewer for recommendation for publication of our study. We appreciate their time and efforts in reviewing our manuscript.

Reviewer #3 (Remarks to the Author):

To my mind, the authors have satisfactorily responded to the referees comments and appropriately modified the manuscript. The paper is a very interesting and original study and should be published as is.

We thank the reviewer for positive assessments on our study and recommendation for publication. We appreciate their time and efforts in reviewing our manuscript.

REVIEWERS' COMMENTS

Reviewer #1 (Remarks to the Author):

I thank the authors for responding to my questions, and all of them have been addressed.